# Translational regulation enhances distinction of cell types in the nervous system

Toshiharu Ichinose[1,2]*, Shu Kondo[3], Mai Kanno[1,2], Yuichi Shichino[4], Mari Mito[4], Shintaro Iwasaki[4,5], Hiromu Tanimoto[2]*

[1]Frontier Research Institute for Interdisciplinary Sciences, Tohoku University, Sendai, Japan; [2]Graduate School of Life Sciences, Tohoku University, Sendai, Japan; [3]Faculty of Advanced Engineering, Tokyo University of Sciences, Tokyo, Japan; [4]RNA Systems Biochemistry Laboratory, RIKEN Cluster for Pioneering Research, Wako, Saitama, Japan; [5]Department of Computational Biology and Medical Sciences, Graduate School of Frontier Sciences, The University of Tokyo, Kashiwa, Japan

*For correspondence:
toshiharu.ichinose.c1@tohoku.ac.jp (TI);
hiromut@m.tohoku.ac.jp (HT)

**Abstract** Multicellular organisms are composed of specialized cell types with distinct proteomes. While recent advances in single-cell transcriptome analyses have revealed differential expression of mRNAs, cellular diversity in translational profiles remains underinvestigated. By performing RNA-seq and Ribo-seq in genetically defined cells in the *Drosophila* brain, we here revealed substantial post-transcriptional regulations that augment the cell-type distinctions at the level of protein expression. Specifically, we found that translational efficiency of proteins fundamental to neuronal functions, such as ion channels and neurotransmitter receptors, was maintained low in glia, leading to their preferential translation in neurons. Notably, distribution of ribosome footprints on these mRNAs exhibited a remarkable bias toward the 5′ leaders in glia. Using transgenic reporter strains, we provide evidence that the small upstream open-reading frames in the 5′ leader confer selective translational suppression in glia. Overall, these findings underscore the profound impact of translational regulation in shaping the proteomics for cell-type distinction and provide new insights into the molecular mechanisms driving cell-type diversity.

## eLife assessment

This **valuable** article explores the role of translational regulation in the establishment of differential gene expression between neurons and glia in *Drosophila*. The article uses Ribo-seq to show extensive variation in the translation efficiency of specific transcripts between neurons and glia. The evidence supporting the model is **solid**, although only one example (that exhibits very strong differential transcriptional expression between one class of neurons and glia) is studied in detail for translation efficiency.

## Introduction

Gene expression is regulated both at the transcription and translation levels (*Becker et al., 2018*; *Casas-Vila et al., 2017*; *Li et al., 2020*; *Liu et al., 2016*; *Schwanhäusser et al., 2011*), and its heterogeneity defines the specialized morphologies and functions of cells. The *Drosophila* brain is a well-studied model tissue with a diverse array of cell types, classifiable by morphology, cell lineage, or gene expression (*Scheffer et al., 2020*; *Zeng and Sanes, 2017*). Recent advances in single-cell transcriptomics have identified groups of differentially expressed genes and provided an in-depth

overview of transcriptional regulations (*Croset et al., 2018*; *Davie et al., 2018*; *Li et al., 2022*). While these inventories provided a powerful way to classify cell types, there have been cases falling short in explaining proteomic or morphological diversity (*Lago-Baldaia et al., 2023*; *Li et al., 2020*). Therefore, post-transcriptional regulations play pivotal roles in distinguishing cell-type-specific proteomes.

Ribosome profiling or Ribo-seq, which is based on deep sequencing of mRNA fragments protected by ribosomes from RNase treatment (ribosome footprints), has been a powerful approach to provide a genome-wide snapshot of protein synthesis ('translatome') (*Ingolia et al., 2009*). Application of this method, combined with transcriptome analysis, revealed multiple layers of translational regulation in cells. For example, this comparison allowed measurements of translational efficiency (TE), which is quantified as the number of ribosome footprints on the coding sequence per mRNA, and discoveries of previously unannotated open-reading frames (ORFs) (*Dunn et al., 2013*; *Ingolia et al., 2011*; *Ingolia et al., 2009*; *Zhang et al., 2018*). While TE profiles have been reported to be variable among dissected animal tissues (*Fujii et al., 2017*; *Wang et al., 2021*; *Zhang et al., 2018*), differences in translational regulations among identified cell types remain unclear.

Applying ribosome profiling to *Drosophila* heads, we here examine the comprehensive landscape of translational profiles between neuronal and glial cells. Due to the size of the fly brain (~0.5 mm) and intricate intercellular adhesions among neurons and glia (*Kremer et al., 2017*), surgical separation is impractical. We thus biochemically purified ribosome-bound mRNAs through genetic tagging of ribosomes in target cells (*Chen and Dickman, 2017*; *Sapkota et al., 2019*; *Scheckel et al., 2020*; *Thomas et al., 2012*; *You et al., 2021*) and further performed Ribo-seq and RNA-seq. By this comparative transcriptome-translatome analyses, we suggest that differential translational programs enhance the distinction of protein synthesis between neuronal and glial cells.

## Results

### Comparative transcriptome-translatome analyses reveal translational suppression of selective groups of proteins in the fly heads

To gain an overview of the translation status, we first applied conventional Ribo-seq in the whole fly head,and successfully monitored footprint distribution at a single-codon resolution (*Figure 1A*, see 'Materials and methods' for technical details). The majority (96.2%) of ribosome footprints was mapped onto the annotated coding sequences (CDS), and its distribution displayed a clear 3-nt periodicity, reflecting the codon-wise movement (*Figure 1B*).

To compare transcriptome and translatome, we also performed RNA-seq from the same lysate (*Figure 1A*). As previously reported, the transcript level and the number of ribosome footprints did not always match, suggesting substantial posttranscriptional regulations ($R^2$ = 0.664; *Figure 1C*). For instance, while *Shaker* (*Sh*) and *Trehalase* (*Treh*), which encode a voltage-gated K$^+$ channel and an enzyme that hydrolyzes trehalose, respectively, were similar regarding transcript levels, far more ribosome footprints were detected on *Treh* (*Figure 1D and E*). We therefore measured TE, ribosome footprints normalized by mRNA reads. TE was much higher for *Treh* than *Sh* (*Figure 1F*), and we found a striking genome-wide variability with more than 20-fold TE difference between the 5 and 95 percentiles (*Figure 1G*). Kyoto Encyclopedia of Genes and Genome (KEGG) pathway enrichment analysis revealed that the transcripts involved in fatty acid metabolism and proteasome are actively translated (*Figure 1H*). In contrast, ribosome proteins, as previously reported (*Chen and Dickman, 2017*; *Cho et al., 2015*), and proteins mediating neuronal ligand–receptor interactions were significantly enriched in the transcripts with low TE, suggesting translational suppression (*Figure 1H*). Indeed, many transcripts encoding ligand- or voltage-gated ion channels, G-protein coupled receptors (GPCR) showed remarkably low TE (*Figure 1C and I*). These results suggest translational regulations specific to neuronal transcripts in the fly head.

### Translational regulation enhances the difference in the gene expression profiles between cell types

Because the translatome/transcriptome status of the whole heads was a mixed average of diverse cell types, such as neurons, glial cells, fat bodies, and muscles, we set up an experimental approach to dissect cell-type-specific translational regulations. By expressing epitope-tagged RpL3 (uL3 in universal nomenclature) (*Chen and Dickman, 2017*) under the control of UAS using the *nSyb-* or the

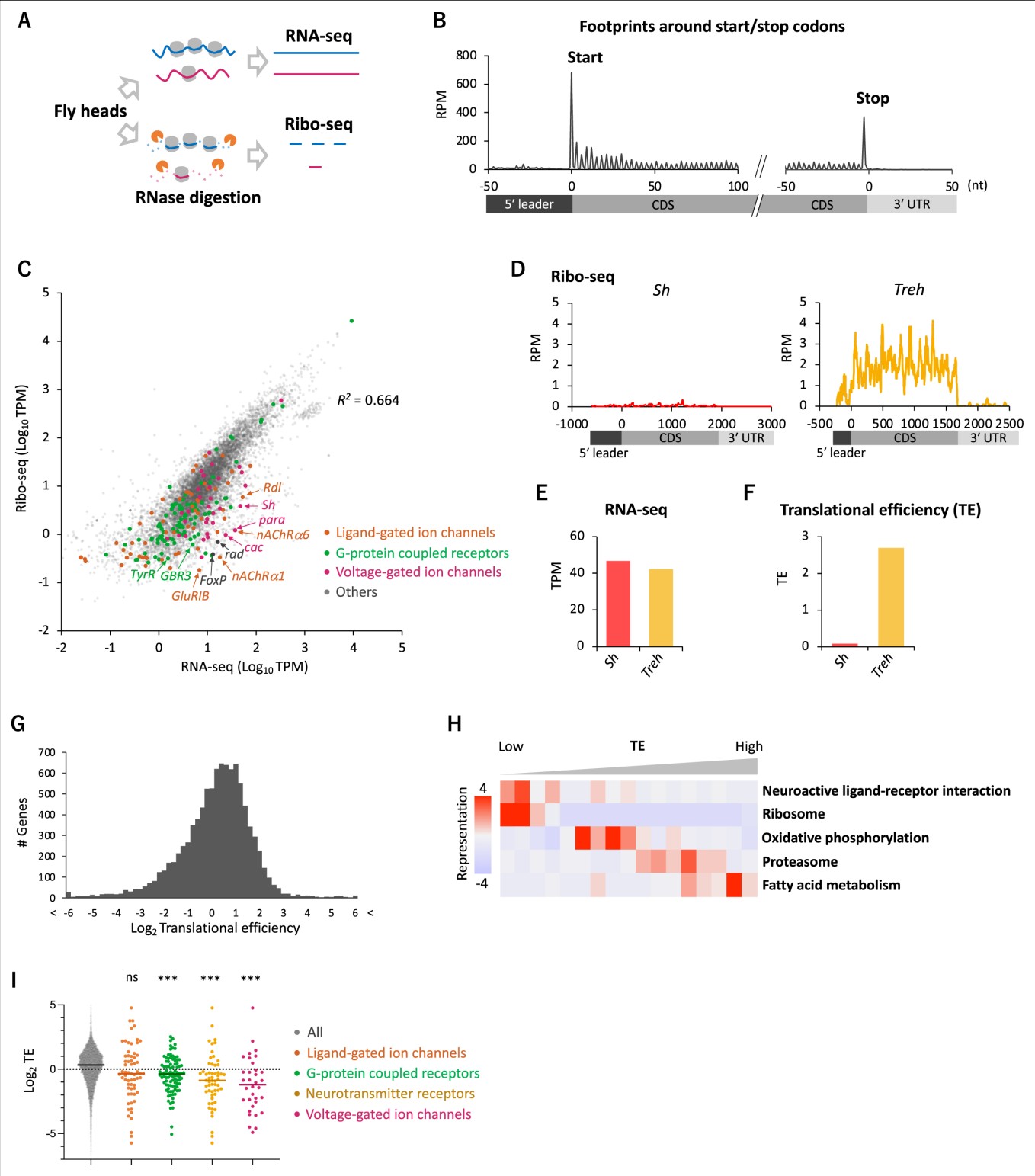

**Figure 1.** Comparative transcriptome-translatome analyses in the *Drosophila* head. (**A**) Schematics. Fly head lysate is digested with RNase I for Ribo-seq, while not for RNA-seq. Resultant short fragments or the whole mRNA are reverse-transcribed and sequenced. (**B**) Meta-genome ribosome distribution (estimated P-sites of the 21-nt fragments), relative to the annotated start and stop codons. RPM: reads per million. (**C**) Scatter plots of mRNA reads (x-axis, TPM: transcripts per million) and ribosome footprints on coding sequences (CDS) (y-axis, TPM). Several neuron-related genes are highlighted with colors and arrows. The squared Pearson's correlation coefficient ($R^2$) is indicated. (**D–F**) Ribosome footprints (**D**), mRNA level (**E**), and translational efficiency (TE) (**F**) of *Shaker-RB* (*Sh*) and *Trehalase-RA* (*Treh*). TE is calculated as ribosome footprints on CDS (TPM) divided by the mRNA

*Figure 1 continued*

level (TPM). (**G**) Histogram of TE. The bin size is 0.2 in the unit of log 2. In total, 9611 genes with at least one read in both Ribo-seq and RNA-seq are plotted. (**H**) Kyoto Encyclopedia of Genes and Genomes (KEGG) pathways enrichment analysis, visualized by iPAGE (*Goodarzi et al., 2009*), based on TE. The 9611 genes are ranked and binned according to TE (left to right: low to high), and over- and under- representation is tested. The presented KEGG pathways show p-values less than 0.0005. (**I**) TE of transcripts in the denoted Gene Ontology terms. Bars represent the median. ns: p>0.05; ***p<0.001; in the Dunn's multiple-comparisons test, compared to the 'all' group.

*repo-GAL4* drivers, we immunopurified the tagged ribosomes and associated mRNAs separately from neurons and glia, and performed Ribo-seq (*Figure 2A*). By immunohistochemistry, we confirmed that *UAS-RpL3::FLAG* on the third chromosome exhibited minimum leakage expression in the brain and did not display any apparent morphological defects upon expression using either driver compared to other insertions or constructs (*Chen and Dickman, 2017*; *Huang et al., 2019*; *Thomas et al., 2012*; *Figure 2A*, *Figure 2—figure supplement 1A and B*). The exogenously expressed RpL3::FLAG was highly concentrated in cell bodies but also detectable in neurites, consistent with the subcellular localization of the endogenous ribosome (*Figure 2—figure supplement 1C and D*).

Through the purification of FLAG-tagged ribosomes, we successfully profiled translatome from neurons and glial cells in the fly heads: footprints were found on 10,821 (78.4% of all the annotated genes) and 10,994 (79.7%) genes in neurons and glia, respectively, with decent reproducibility among the biological replicates ($R^2 > 0.9$, *Figure 2—figure supplement 2A*). The FLAG-tagged RpL3 in the corresponding cells far exceeded the endogenous RpL3, as RpL3 reads were 7.8 and 42.7 times higher in neurons and glia, respectively, compared to the wild-type whole-head samples (*Figure 2—figure supplement 2B*). The known marker genes were strongly enriched while non-target markers were depleted (*Figure 2B*, *Figure 2—figure supplement 2C and D*; *Croset et al., 2018*; *Davie et al., 2018*; *Li et al., 2022*), and the KEGG enrichment analysis showed significant enrichment of footprints on genes associated with the known functions of these cell types (*Figure 2—figure supplement 2E*). Interestingly, the KEGG analysis also revealed that neurons exhibit a greater extent of protein synthesis related to oxidative phosphorylation and mitochondrial ribosome proteins, while glial cells show higher expression of proteins associated with glycolysis (*Figure 2—figure supplement 2E and F*). These findings support the glia-neuron lactate shuttle hypothesis, a recently proposed concept of metabolic specialization (*Mason, 2017*; *Volkenhoff et al., 2015*). Furthermore, apart from the annotated CDS, we detected clustered ribosome footprints on *Hsr-ω*, previously annotated as a long non-coding RNA, strongly suggesting the synthesis of hitherto undescribed polypeptides (*Figure 2—figure supplement 2G*; *Singh, 2022*). Altogether, the combination of genetic labeling of ribosomes in selective cell types and Ribo-seq revealed the differential translatome profiles in the fly heads.

To further examine translational regulation by calculating TE, we performed RNA-seq from the same immunoprecipitated complexes, similar to Translating Ribosome Affinity Purification (TRAP) (*Heiman et al., 2008*; *Figure 2A*, *Figure 2—figure supplement 3A and B*). Because this approach relies on the 80S-ribosome-mRNA complex, we may miss mRNA with little or no translation. Nevertheless, our transcriptome was similar to the sn-transcriptome data (*Li et al., 2022*; *Figure 2—figure supplement 3C*). We identified groups of genes undergoing neuron- or glia-specific translational regulations compared to the whole heads (*Figure 2—figure supplement 4A*). Genes mediating fatty acid metabolism and degradation, for example, were actively translated in the whole head, but showed lower TE in neurons or in glia (*Figures 1H and 2C*). Because many of these genes are highly expressed in the fat bodies (*Dobson et al., 2018*), these results suggest selective translational enhancement in the fat body. Strikingly, TE of genes involved in neuroactive ligand–receptor interaction was significantly higher in neurons but lower in glia (*Figure 2C and D*), suggesting cell-type-specific translational regulation of these genes.

This differential translational regulation was highlighted in the weak TE correlation between neurons and glia ($R^2 = 0.534$, *Figure 2E*). We found a genome-wide tendency that genes transcribed less in glia are further suppressed at translation (*Figure 2F*). Specifically, many functionally characterized neuronal genes, such as voltage- or ligand-gated ion channels, G-protein-coupled receptors, neuropeptides, and proteins for visual perception, showed particularly lower TE in glia (*Figure 2E, G, and H*. *Figure 2—figure supplement 4C*). For these genes, the distinction between neuronal and glial cells was much exaggerated at the level of translation than at transcription (*Figure 2H*). Consistently on the genome-wide scale, the inter-cell-type correlation became weaker in Ribo-seq data

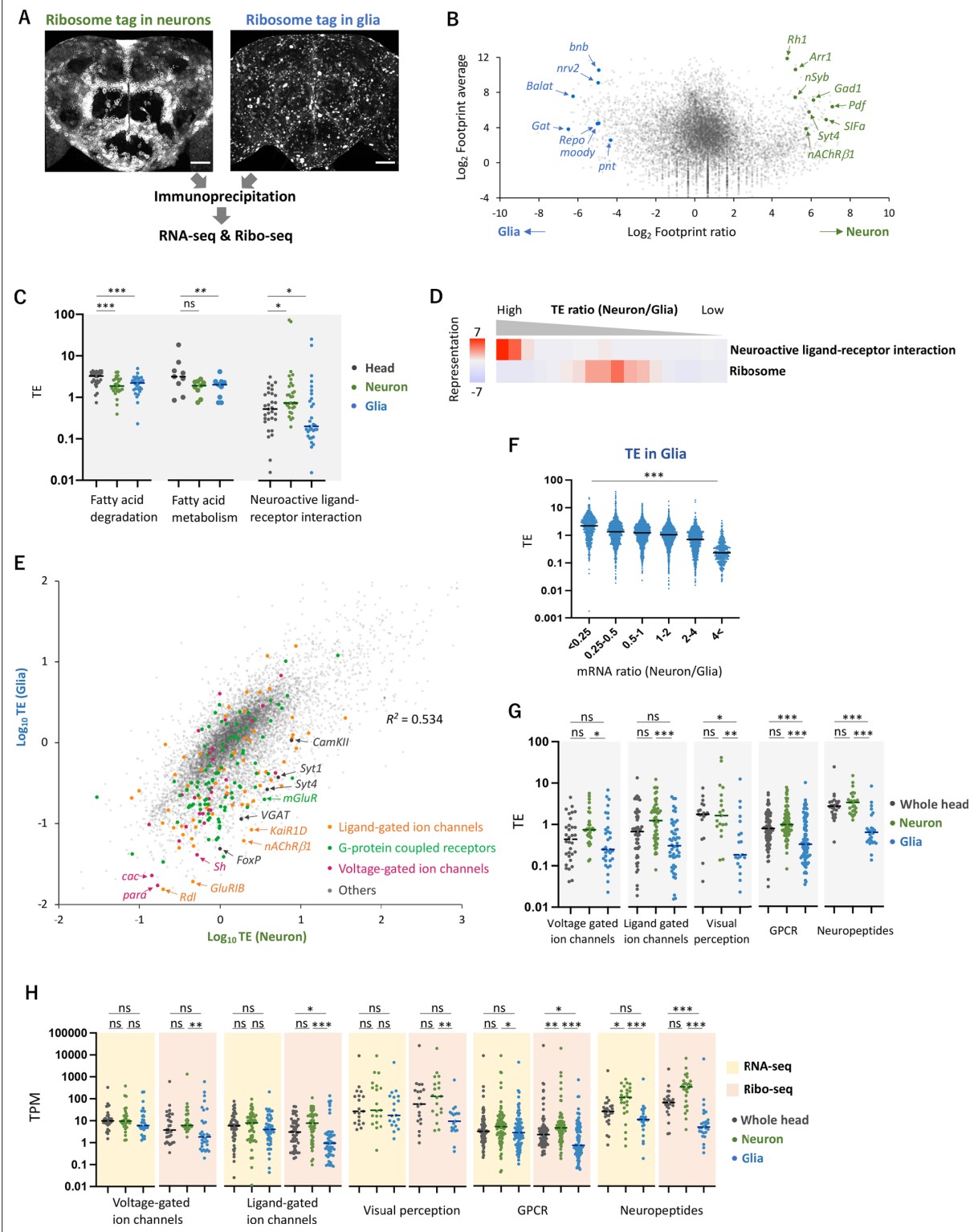

**Figure 2.** Cell-type-specific Ribo-seq and RNA-seq reveal differential translational regulations. (**A**) Schematics. FLAG-tagged ribosome protein L3 (RpL3::FLAG) is expressed in neurons (*nSyb-GAL4*) or in glial cells (*repo-GAL4*). RNA-seq and Ribo-seq are performed following immunoprecipitation. Whole brain images of the exogenously expressed RpL3::FLAG are shown. Scale bars: 50 µm. (**B**) The MA plot of ribosome footprints on coding sequences (CDS) among neurons and glia. Each gene is plotted according to the fold change (x-axis) and the average (y-axis) in the unit of $\log_2$. Several marker genes are highlighted with green (neuron) or blue (glia). (**C**) Translational efficiency (TE) of genes in the denoted Kyoto Encyclopedia of Genes and Genomes (KEGG) pathways in the whole head (black), neurons (green), or in glia (blue). Genes with transcripts per million (TPM) > 1 in the RNA-

*Figure 2 continued on next page*

Figure 2 continued

seq dataset are plotted. Bars represent the median. *p<0.05, **p<0.01, ***p<0.001, Dunn's multiple-comparisons test. (**D**) KEGG pathway enrichment analysis based on the ratio of TE in neurons to in glia. All genes with at least one read in both cell types (total 9732 genes) are ranked and binned according to the neuron-to-glia ratio (left to right: high to low), and over- and under-representation is tested. The presented KEGG pathways show p-values less than 0.0005. (**E**) Scatter plot of TE in neurons (x-axis) and in glia (y-axis). The squared Pearson's correlation coefficient ($R^2$) is indicated. (**F**) TE in glia plotted according to the ratio of mRNA expression in neurons compared to glia. ***p<0.001, Kruskal–Wallis test. All the 7933 genes showing TPM > 1 in RNA-seq are analyzed. (**G**) TE of transcripts, showing at least one read, in the indicated Gene Ontology (GO) terms. Bars represent the median. **p<0.01, ***p<0.001, Dunn's multiple-comparisons test. (**H**) Read counts of genes (TPM) in the indicated GO terms in RNA-seq (yellow) and in Ribo-seq (pink). The gray, green, and blue dots indicate the read counts in the whole head, neurons, and glial cells, respectively. ns: p>0.05, **p<0.01, ***p<0.001, Dunn's multiple-comparisons test.

The online version of this article includes the following figure supplement(s) for figure 2:

**Figure supplement 1.** Cell-type-specific ribosome profiling.

**Figure supplement 2.** Cell-type-specific ribosome profiling.

**Figure supplement 3.** Cell-type-specific RNA-seq.

**Figure supplement 4.** Translational efficiency in neurons, glial cells, and the whole heads.

compared to in RNA-seq ($R^2$ = 0.59 vs. 0.81, *Figure 2—figure supplement 3B*). Altogether, these data indicate substantial contributions of translational regulation to shaping the cell-type-specific protein expression.

## Biased distribution of ribosomes toward upstream ORFs of neural genes in glial cells

We next analyzed the distribution of ribosome footprints on the differentially translated transcripts (DTT). Fat-body-related genes showed lower TE in neurons compared to the whole head (*Figure 2C*). Among these genes, we found a remarkable ribosomal accumulation on the start codon specifically in neurons (*Figure 3—figure supplement 1A and B*), as if the first round of the elongation cycle was arrested in neurons. Through the reanalysis of the published RNA-seq data (*Dobson et al., 2018*), we found that mRNAs showing strong ribosomal accumulation on the start codons are highly abundant in the fat bodies (*Figure 3—figure supplement 1C*). On the other hand, DTTs suppressed in glial cells compared to neurons (defined as genes with more than 10 times higher TE in neurons than in glia, n = 161), we noticed that glial ribosome footprints were remarkably biased toward 5′ leaders (*Figure 3A and B*). Notably, this pattern was not obvious on the genome-wide scale (*Figure 3A and B*). The high 5′ leader/CDS ratio of ribosome footprints in glia was commonly observed on many transcripts with known neuronal functions, such as *Rab3*, *Syt4*, *Arr1*, and *Syn* (*Figure 3D and E*). Conversely, we observed accumulated ribosome footprints on the 5′ leaders of several glial marker genes specifically in neurons (*Figure 3—figure supplement 2*). Altogether, these results suggest that the translation of 5′ leaders in selective mRNAs differentiates protein synthesis among cell types.

We reasoned translational downregulation via upstream ORFs (uORFs) in the 5′ leaders in glia, as the translation of uORFs was reported to suppress that of the downstream main ORF (*Ferreira et al., 2013*; *Zhang et al., 2019*; *Zhang et al., 2018*). Consistent with this idea, metagene plot around the AUG codons on 5′ leaders revealed strong accumulation of footprints on the upstream AUG codons, similar to those observed on the initiation codon of CDSs (*Figure 4A and B*). We calculated the footprint accumulation score on each codon (defined as the ratio of footprints on each codon with surrounding –50/+50 nt), and found that upstream AUG and the near cognate codons (NUG or AUN) showed relatively high accumulation (*Figure 4C*). On the other hand, inside the annotated CDS, none of the codons exhibited such significant accumulation (*Figure 4D*). Consistently, we found that transcripts related to neuronal functions typically contain long 5′ leaders and many upstream AUG (*Figure 4—figure supplement 1*). We thus propose that glial cells suppress the translation of neuronal transcripts by stalling ribosomes on 5′ leader via uORF.

## uORFs in *Rh1* confer translational suppression in glia

We next asked whether the 5′ leader sequences of neuronal genes cause cell-type differences in translation. To this end, we focused on *Rh1* (*Rhodopsin 1*, also known as *ninaE*), which encodes an opsin, also detecting stimuli of other sensory modalities (*Leung et al., 2020*; *O'Tousa et al., 1985*;

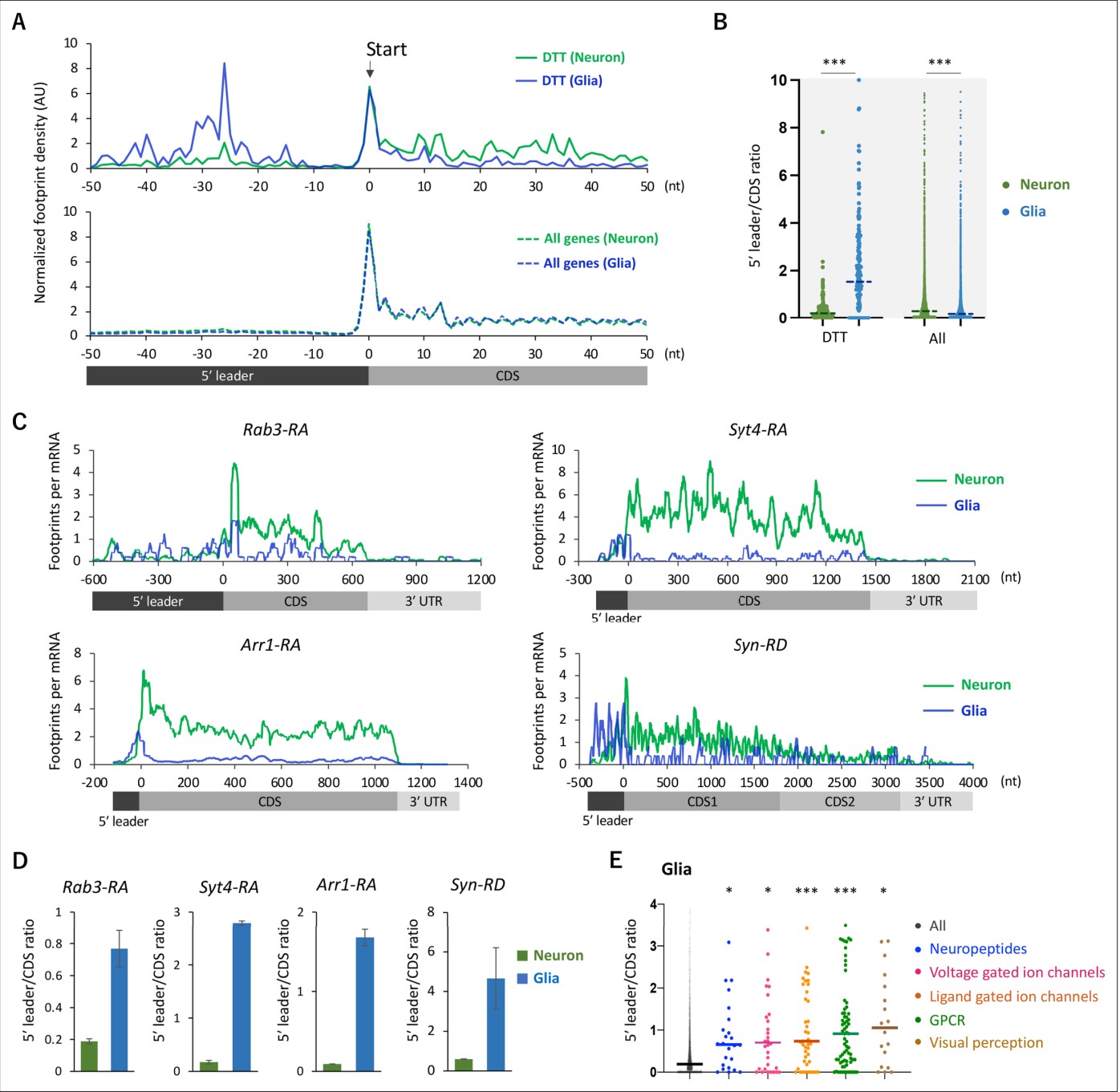

**Figure 3.** Ribosome stalling on the 5′ leaders of differentially translated transcripts (DTTs) in glia. (**A**) Ribosome distribution (estimated P-sites) on the 161 DTTs around the start codons (solid lines; start ±50 nt). These DTTs are defined as transcripts showing more than 10 times higher translational efficiency (TE) in neurons compared to glia. The dotted lines in the bottom graph indicate the genome-wide distribution. All the transcripts showing transcripts per million (TPM) > 1 in RNA-seq both in neurons and glia are considered (7933 genes in total), and the height is normalized by the total reads on this region. (**B**) Ratio of ribosome density on 5′ leader (TPM) to coding sequences (CDS) (TPM) of the 161 DTTs or of all transcripts in neurons (green) or in glia (blue). The bars represent the median. ***p<0.001, Mann–Whitney test of ranks. (**C**) Distribution of ribosome footprints on the representative neuronal transcripts. Ribosome footprints (reads per million [RPM]) normalized by the mRNA level (TPM) are shown. Note that *Syn-RD* harbors a stop codon in the CDS but a fraction of ribosomes skip it, generating two annotated open-reading frames (ORFs) (CDS1 and CDS2) (**Klagges et al., 1996**). (**D**) Ratio of ribosome density on 5′ leader to CDS (mean ± standard error of mean of the biological replicates). (**E**) Ratio of ribosome density on 5′ leader to CDS on transcripts in the indicated Gene Ontology (GO) terms in glia. *p<0.05, ***p<0.001, Dunn's multiple-comparisons test compared to the 'all' group.

The online version of this article includes the following figure supplement(s) for figure 3:

*Figure 3 continued on next page*

**Figure supplement 2.** Translational suppression of glial marker genes in neurons.

*Shen et al., 2011*; *Zuker et al., 1985*). Consistently, active translation of Rh1 was almost exclusively observed in neurons (*Figure 5A*). Similar to other neuronal genes shown in *Figure 3C*, the distribution of ribosome footprints was distinct among neuronal and glial cells: they were heavily biased to 5′ leader in glia, with the striking accumulation on the putative uORFs composed only of the start and stop codons (*Figure 5B and C*).

To address the function of these sequences on differential translation, we constructed a transgenic reporter strain using the *Rh1* UTR sequences under the control of UAS (*Figure 5D*), and directed gene expression ubiquitously using *Tub-GAL4*. While the reporter mRNA was detected both in neuronal and glial cells, the protein levels were much more heterogeneous and strikingly weak in glia (*Figure 5E*, *Figure 5—figure supplement 1A*). The control reporter strain (*DeLuca and Spradling, 2018*), on the other hand, exhibited more ubiquitous expression, with significantly higher fluorescent intensity in glia (*Figure 5E and F*). Driving the reporter expression using the *nSyb-* or *repo-GAL4* further corroborated cell-type-specific suppression in glia (*Figure 5—figure supplement 1B and C*). Strikingly, when the six-base putative uORFs were mutated, the in vivo protein-to-mRNA ratio of the reporter was significantly increased in glia but not in neurons (*Figure 5G and H*). Based on these results, we propose that glial cells selectively suppress the protein synthesis of neuronal genes through uORF and thereby enhance the translatome distinction from neurons.

## Discussion

In this study, the comparative translatome-transcriptome analyses in the whole heads, neurons, and glial cells revealed the significant diversity of translational regulations across different cell types.

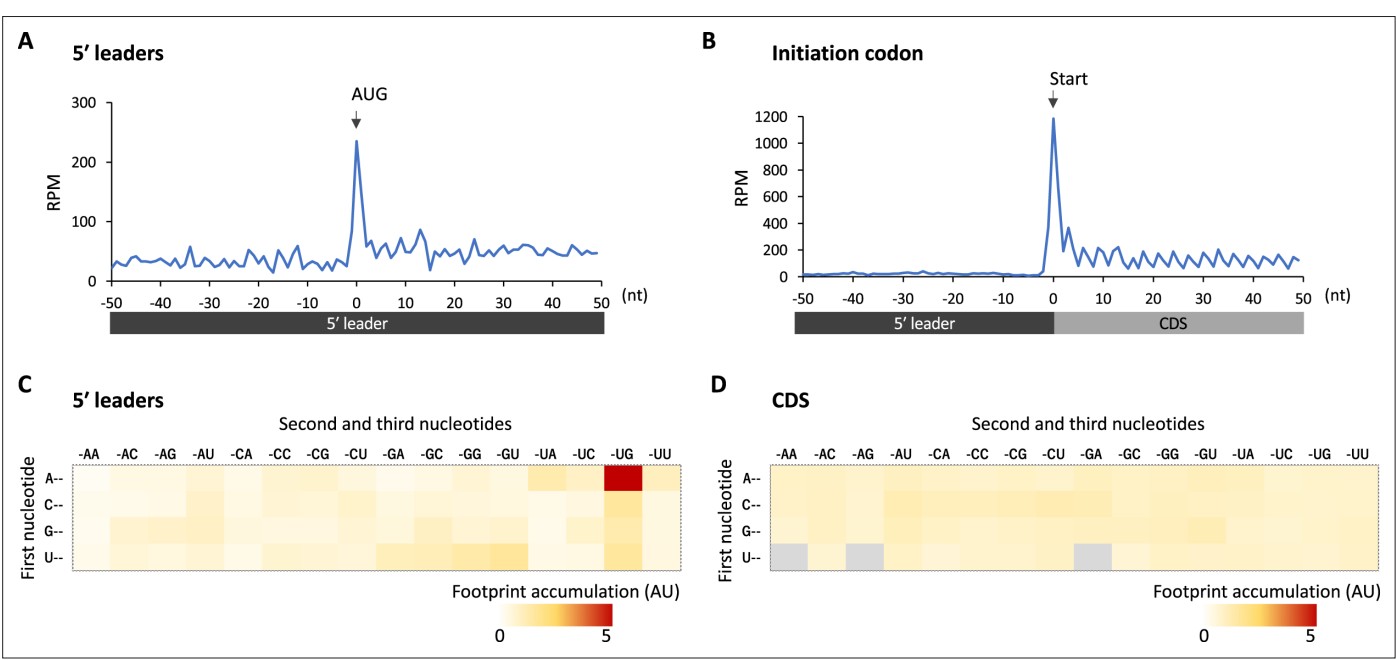

**Figure 4.** Footprint accumulation on upstream AUG in glia. (**A**) Meta-genome ribosome distribution (estimated P-sites of the 32-nt fragments) around the upstream AUG codons in glia. (**B**) Meta-genome ribosome distribution (estimated P-sites of the 32-nt fragments) around the annotated start codons in glia. (**C**) Footprint accumulation on 5′ leader in glia, defined as the number of ribosome footprints (estimated P-sites) on each codon normalized by the average on the surrounding (−50 to +50) regions. (**D**) Footprint accumulation inside the annotated coding sequences (CDS) in glia. Annotated in-frame codons except the start and the stop codons are considered. AU: arbitrary unit.

The online version of this article includes the following figure supplement(s) for figure 4:

**Figure supplement 1.** Neuronal transcripts harbor long 5′ UTR containing numerous upstream open-reading frames (uORFs).

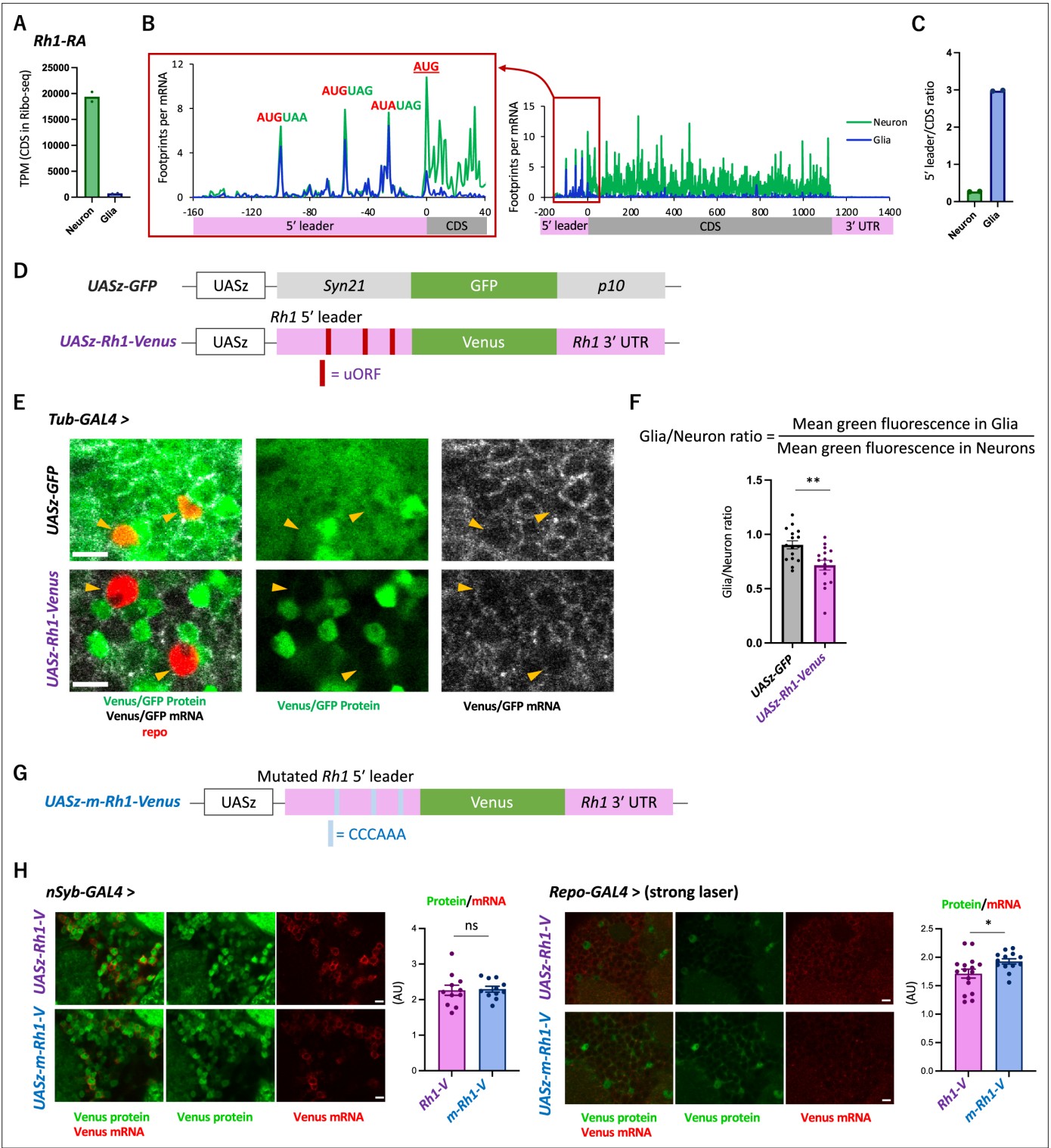

**Figure 5.** The transgenic *Rh1-Venus* reporter reveals differential translation in neuronal and glial cells. (**A**) Reads on coding sequences (CDS) of *Rh1-RA* in Ribo-seq. (**B**) Ribosome distribution (estimated P-sites) on *Rh1-RA* in neurons (green) and in glia (blue), with 0 on the x-axis indicating the start codon of the CDS. Six-base upstream open-reading frames (ORFs), consisting of consecutive start (or the near-cognate) and stop codons, are highlighted. Note that footprints are normalized by the mRNA level (transcripts per million [TPM]). (**C**) Ratio of ribosome density on 5' leader (TPM) to CDS (TPM) in neurons (green) or in glia (blue). The bars and the dots represent the median and individual data points, respectively. (**D**) Schematics of the control (*UASz-GFP*) or the *Rh1* (*UASz-Rh1-Venus*) reporter. For the *Rh1* reporter, 5' leader and 3' UTR sequences of *Rh1-RA* are fused to CDS of the *Venus* fluorescent protein. For the control reporter, synthetic 5' leader sequences (*syn21*) and viral *p10* terminator are fused to GFP (**DeLuca and Spradling,**

*Figure 5 continued on next page*

*Figure 5 continued*

*2018*). Note that both reporters contain the same promoter (*UASz*) (*DeLuca and Spradling, 2018*) and are inserted onto the identical genomic locus (*attP40*). (**E**) Expression of the *Rh1*- or the control reporters driven by *Tubulin-GAL4*. Sliced confocal images of the cortical regions next to the antennal lobe are shown. Green: EGFP or Venus fluorescent signal. Red: immunohistochemical signal of repo protein as a glial marker. Gray: EGFP or Venus mRNA. Orange arrowheads indicate glial cells marked by the repo expression. Scale bars: 5 µm. (**F**) Quantification of the green fluorescent intensity in glial nuclei, normalized by the fluorescence in neurons. Glial intensity was measured as mean intensity in the repo-positive pixels, and was normalized by the mean intensity in the repo-negative pixels. **p<0.01, Mann–Whitney test of ranks. (**G**) Schematics of the mutated *Rh1* reporter (m-Rh1). The minimal upstream open-reading frame (uORF) is replaced with CCCAAA. (**H**) The expression of the *Rh1*- or *m-Rh1*- reporters, driven by the *nSyb*- or the *repo*-GAL4. Sliced confocal images of the cortical regions next to the antennal lobe are shown. Scale bars: 5 µm. Green: Venus fluorescent signal. Red: Venus mRNA signal. The total protein signal was normalized by the total mRNA signal for each brain. N = 8 (*nSub>Rh1*), 8 (*nSyb>m-Rh1*), 16 (*repo>Rh1*), 13 (*repo>m-Rh1*). ns: p>0.05, *p<0.05, Mann–Whitney test of ranks.

The online version of this article includes the following figure supplement(s) for figure 5:

**Figure supplement 1.** The *Rh1* reporter expression in neurons or in glia.

Particularly noteworthy was the differential translation of transcripts encoding neuronal proteins, including ion channels and neurotransmitter receptors (*Figure 2*). These neuronal transcripts exhibited preferential translation in neurons (*Figure 2*), and the relatively long 5′ UTR of these transcripts strongly stalled ribosomes in glia (*Figures 3 and 4*, *Figure 4—figure supplement 1*). This characteristic feature of long 5′ leaders containing numerous uORFs is also observed in neuronal transcripts in mammals (*Glock et al., 2021*). While the 5′ leader-mediated translational regulations are known to be critical for quick response to environmental changes, such as starvation or oxidative stress (*Harding et al., 2003*; *Mueller and Hinnebusch, 1986*; *Young and Wek, 2016*), our study sheds light on its roles in contrasting protein expression among cell types. Furthermore, considering a pivotal role of de novo protein synthesis for long-lasting adaptation (*Flexner et al., 1963*; *Tully et al., 1994*), it is plausible that similar mechanisms are employed for neuronal plasticity as well.

The next obvious question would be how the translation of neuronal transcripts is differentiated among neuronal and glial cells. Ribosomes can initiate translation at uORFs more frequently in glia. Alternatively, the post-termination 40S subunits reinitiate translation of the coding sequence more often in neurons. These two possibilities can be distinguished by profiling the rates of initiation or reinitiation, achievable through sequencing the footprints of the 40S subunits, known as translation complex profile sequencing, coupled with conventional Ribo-seq (*Archer et al., 2016*; *Bohlen et al., 2020*; *Wagner et al., 2020*). Although there are various technical challenges, the application of this technique to specific cells within the brain would elucidate these possibilities. Furthermore, previous studies have identified eIF1, eIF2α kinases, and DENR/MCT1 as facilitators of translation of main ORF whose 5′ leaders harbor uORFs (*Ivanov et al., 2010*; *Schleich et al., 2014*; *Sonenberg and Hinnebusch, 2009*; *Zhou et al., 2020*). Interestingly, all these proteins are expressed more in neurons than in glia in our dataset (*Supplementary file 1*). Selective activation of these molecular machineries might underlie the cell-type-specific translation.

Our cell-type-specific translatome analysis further revealed translational regulations beyond 5′ leaders. We found a remarkable ribosomal stall at the initiation codon in several transcripts, a phenomenon observed in neurons but not in the entire heads (*Figure 3—figure supplement 1*). These transcripts are known to be massively expressed in the fat bodies but less in the nervous system (*Figure 3—figure supplement 1C*; *Dobson et al., 2018*), and the translation was further suppressed in neurons (*Figure 3—figure supplement 1*). Therefore, transition from initiation to elongation may serve as another regulatory checkpoint of protein synthesis (*Harnett et al., 2022*; *Wang et al., 2019*), which enhances cell-type distinctions. Furthermore, we found ribosome footprints also on the 3′ UTR of certain transcripts, such as *Synapsin* (*Figure 3C*). Stop-codon readthrough has been reported to be more frequent in neurons than in other cell types (*Hudson et al., 2021*; *Karki et al., 2022*; *Prieto-Godino et al., 2016*). Because the readthrough events extend the protein C-terminus, its regulation can add yet another layer of cell-type diversity (*Dunn et al., 2013*; *Jungreis et al., 2011*; *Klagges et al., 1996*). Altogether, we here propose that translational regulations further differentiate transcriptome distinctions, thereby shaping the cellular identity.

Due to the specialized functions of neuronal and glial cells, they express distinct sets of proteins. Neurons allocate more ribosomes to proteins related to neurotransmission, visual sensing, and oxidative phosphorylation, while glial cells synthesize transporters and enzymes for metabolism of amino

acid, fatty acid, or carbohydrates (*Figure 2—figure supplement 1F*). Despite these clear differences and specialization, a significant amount of neuronal and glial cells has a common developmental origin. They originate from a stem cell lineage known as neuro-glioblasts (*Lai and Lee, 2006*; *Viktorin et al., 2011*), and the fate of these cells can be altered by the expression of a single gene, *glial cells missing* (*gcm*) (*Hartenstein, 2011*; *Hosoya et al., 1995*). Therefore, translational regulations, in addition to transcriptional diversity, may play a particularly important role in these sister cell types with distinct physiological roles.

In the *Drosophila* brain, approximately 100 stem cell lineages diverge into more than 5000 morphologically distinct cell types (*Ito et al., 2013*; *Scheffer et al., 2020*; *Yu et al., 2013*). Hence, translational regulations similar to those described in this study, or other possible regulations, may play significant roles in further differentiating neuronal or glial subtypes. Consistent with this idea, our *UAS-Rh1-Venus* reporter showed heterogeneous expression even among neurons, contrasting with the more uniform expression observed in the control *UASz-GFP* reporter (*Figure 5E*, *Figure 5—figure supplement 1B and C*). In accordance, choline acetyltransferase (ChAT), an enzyme needed to synthesize acetylcholine, and vesicular acetylcholine transporter (VAChT) are transcribed in many glutamatergic and GABAergic neurons but its protein synthesis is inhibited (*Chen et al., 2023*; *Lacin et al., 2019*). Substantial post-transcriptional regulations are also implicated during development (*Li et al., 2020*; *Zhang et al., 2016*). Taken together, multiple layers of transcriptional and post-transcriptional regulations should shape the proteome diversity of cell types in the nervous system. Further comparative transcriptome-translatome analyses using more specific GAL4 drivers should highlight the diversity of translational regulations leveraged in the brain.

## Limitation of the study

Because our cell-type-specific Ribo-seq and RNA-seq are based on immunoprecipitation of genetically tagged RpL3 (*Chen and Dickman, 2017*), the read counts could contain biases, such as underestimation of mRNA level with little or no translational activity, or over- and under-representation of certain cell types originating from the heterogeneous expression of the drivers.

# Materials and methods

## Fly culture and genetics

The flies were reared in a mass culture at 24°C under the 12–12 hr light-dark cycles on the standard cornmeal food. The *Canton-S* strain was used as the wild-type. We utilized the following transgenic strains: *w^1118^;;GMR57C10-GAL4* (*nSyb-GAL4*; BDSC #39171), *w^1118^;;repo-GAL4* (BDSC #7415), *y^1^w^1118^;;tublin-GAL4* (BDSC #5138), *w^1118^;MB010B* (BDSC #68293), *w^1118^;;UAS-RpL3::FLAG* (BDSC #77132) (*Chen and Dickman, 2017*), *y^1^v^1^;UAS-Rh1-Venus* (made in this study; see below), *y^1^v^1^;UAS-m-Rh1-Venus* (made in this study), *w;UASz-GFP* (a kind gift from Dr. Steven DeLuca) (*DeLuca and Spradling, 2018*). Females of the GAL4 drivers were crossed to males of the UAS effectors, and the F1 progenies were used for the experiments. Of note, although *UAS-EGFP::RpL10Ab* (*Thomas et al., 2012*) has been used to isolate ribosomes from specific cells, its expression using the *repo-GAL4* caused lethality in our hands.

## Library preparation for ribosome profiling

### Tissue collection and lysate preparation

Here, 4- to 8-day-old flies with mixed gender were flash-frozen with liquid nitrogen, thoroughly vortexed, and the heads were isolated from the bodies with metal mesh in a similar manner reported previously (*Sun et al., 2020*). Approximately 500 frozen heads were mixed with 400 μl of frozen droplets of lysis buffer (20 mM Tris–HCl pH 7.5, 150 mM NaCl, 5 mM MgCl$_2$, 1 mM dithiothreitol, 1% Triton X-100, 100 μg/ml chloramphenicol, and 100 μg/ml cycloheximide) in a pre-chilled container, then pulverized with grinding at 3000 rpm for 15 s using a Multi-beads Shocker (YASUI KIKAI). Cycloheximide and chloramphenicol were added to the lysis buffer to prevent possible elongation and run-off of cytosolic and mitochondrial ribosomes, respectively. The lysate was slowly thawed at 4°C and the supernatant was recovered after spinning down by a table-top micro centrifuge. The lysate was treated with 10 U of Turbo DNase (Thermo Fisher Scientific) on ice for 10 min to digest the genome DNA. The supernatant was further clarified by spinning at 20,000 × *g* for 10 min.

## Immunoprecipitation

Anti-FLAG M2 antibody (F1804, Sigma-Aldrich) and Dynabeads M-280 bound to anti-mouse IgG antibody (11201D, Invitrogen) were used for immunoprecipitation. 25 µl of the beads solution, washed twice with the aforementioned lysis buffer, was mixed with 2.5 µl of the M2 antibody, and incubated at 4°C for 1 hr with rotation. Beads were incubated with the lysate at 4°C for 1 hr with rotation and washed four times with the lysis buffer. The ribosome-bound mRNA was eluted with 50 µl of 100 µg/ml 3×FLAG peptide (GEN-3XFLAG-25, Protein Ark) dissolved in the lysis buffer.

## RNase digestion and library preparation

Ribosome profiling was performed as described previously (*McGlincy and Ingolia, 2017*; *Mito et al., 2020*) with modifications. We used RNase I from *Escherichia coli* (N6901K, Epicentre) to digest the crude (*Figure 1*, *Figure 2—figure supplement 1C and D*) or the immunoprecipitated (*Figure 2*) lysate. Concentration of RNA in lysate was measured with Qubit RNA HS kit (Q32852, Thermo Fisher Scientific). RNase I was added at a dose of 0.25 U per 1 µg RNA in a 50 µl reaction mixture, which was incubated at 25°C for 45 min. We used 1.36 µg and 0.5 µg RNA to prepare the whole head libraries (*Figure 1*, *Figure 2—figure supplement 1C and D*) and the cell-type-specific libraries (*Figures 2 and 3*), respectively. The RNase digestion was stopped by adding 20 U of SUPERase•In (AM2694, Thermo Fisher Scientific). Ribosomes were isolated by MicroSpin S-400 HR columns (27-5140-01, GE Healthcare). Subsequently, we purified RNA using the TRIzol-LS (10296010, Thermo Fisher Scientific) and Direct-zol RNA Microprep kit (R2062, Zymo Research), and isolated the RNA fragment ranging 17–34 nt by polyacrylamide gel electrophoresis.

The isolated RNA fragments were ligated to custom-made preadenylated linkers containing unique molecular identifiers and barcodes for library pooling, using T4 RNA ligase 2, truncated KQ (M0373L, New England Biolabs) (*Mito et al., 2020*). Ribosomal RNA was depleted by hybridizing to the custom-made biotinylated 2′-O-methyl oligonucleotides with complementary sequences to the *Drosophila* rRNA (see *Supplementary file 3* for the sequences), which can be pulled down using the streptavidin-coated beads (65001, Thermo Fisher Scientific). The rRNA-depleted samples were reverse-transcribed with ProtoScript II (M0368L, New England Biolabs), circularized with CircLigase II (CL9025K, Epicentre), and PCR-amplified using Phusion polymerase (M0530S, New England Biolabs) (*Mito et al., 2020*). The libraries were sequenced with the Illumina HiSeq 4000 system (Illumina) with single-end reads of 50 bases.

## Library preparation for transcriptome analysis

The crude (*Figure 1*) or the immunoprecipitated (*Figure 2*) lysate was prepared using the same protocol as described above, but without RNase digestion. RNA was purified using TRIzol-LS. The libraries were constructed in Azenta Japan Corporation, using the NEBNext Poly(A) mRNA Magnetic Isolation Module (E7760, New England Biolabs) and MGIEasy RNA Directional Library Prep kit (1000006386, MGI tech). Briefly, poly-A tailed mRNAs were enriched with the oligo dT beads, fragmented, and reverse-transcribed using random primers. After the second strand cDNA was synthesized, an adapter sequence was added. DNA library was PCR-amplified. The libraries were sequenced with DNB-seq (MGI tech) with an option of paired end reads for 150 bases.

## Data analysis

Adaptor sequences were removed using Fastp (*Chen et al., 2018*), and the reads that matched to the non-coding RNA were discarded. The remaining reads were mapped onto the *Drosophila melanogaster* release 6 genome. Mapping was performed using STAR (*Dobin et al., 2013*). PCR-duplicated reads were removed by referring to the unique molecular identifiers. The number of uniquely mapped reads are as follows:

## Ribo-seq:

| Sample | #Reads |
| --- | --- |
| *Canton-S*, whole heads | 1,427,090 |

*Continued on next page*

*Continued*

| Sample | #Reads |
|---|---|
| *nSyb-GAL4/UAS-RpL3::FLAG*, after IP, replicate 1 | 2,443,467 |
| *nSyb-GAL4/UAS-RpL3::FLAG*, after IP, replicate 2 | 1,135,284 |
| *repo-GAL4/UAS-RpL3::FLAG*, after IP, replicate 1 | 2,698,259 |
| *repo-GAL4/UAS-RpL3::FLAG*, after IP, replicate 2 | 2,133,920 |
| *nSyb-GAL4/UAS-RpL3::FLAG*, whole heads, replicate 1 | 2,147,092 |
| *nSyb-GAL4/UAS-RpL3::FLAG*, whole heads, replicate 2 | 2,361,588 |
| *repo-GAL4/UAS-RpL3::FLAG*, whole heads, replicate 1 | 1,372,502 |
| *repo-GAL4/UAS-RpL3::FLAG*, whole heads, replicate 2 | 1,846,122 |

RNA-seq:

| Sample | #Reads |
|---|---|
| *Canton-S*, whole heads | 29,132,939 |
| *nSyb-GAL4/UAS-RpL3::FLAG*, after IP, replicate 1 | 40,638,850 |
| *nSyb-GAL4/UAS-RpL3::FLAG*, after IP, replicate 2 | 34,247,848 |
| *repo-GAL4/UAS-RpL3::FLAG*, after IP, replicate 1 | 30,906,375 |
| *repo-GAL4/UAS-RpL3::FLAG*, after IP, replicate 2 | 31,928,337 |

For Ribo-seq analysis, fragments ranging from 20 to 34 nt for whole head samples and 21–36 nt for immunoprecipitated samples were used. For the whole head samples, the position of the P site was estimated as 12 or 13 nt downstream from the 5′ end, for the 20–31 nt or 32–34 nt fragments, respectively (*Ingolia et al., 2009*). For the immunoprecipitated samples, it was estimated as 12 or 13 nt downstream for the 21 nt or 22–36 nt fragments, respectively. Footprints were considered to be on the CDS if the estimated P site was between the annotated start and stop codons. RNA-seq analysis included all fragments greater than 30 nt in length. For genes with alternatively spliced transcripts, the isoform with the highest TPM in the wild-type RNA-seq sample (*Figure 1*) was selected as the 'representative' isoform. If not specified, only the representative isoforms were considered. TE was calculated as TPM of ribosome footprints on CDS divided by TPM of RNA-seq.

The KEGG-enrichment analyses were performed using iPAGE (*Figures 1H and 2D*; *Goodarzi et al., 2009*) or DAVID (*Figure 2—figure supplements 2E* and *Figure 2—figure supplement 4A*; *Dennis et al., 2003*). Statistical tests were performed with GraphPad Prism 9.

For the Fly Cell Atlas data (*Figure 2—figure supplement 3C*), expression level was calculated as the mean RPKM in all cells annotated as neuronal or glial cells in heads (*Li et al., 2022*).

## Reporter construct and the transgenic strain

DNA fragments containing a minimal hsp70 promoter (hsp70Bb) (*DeLuca and Spradling, 2018*), the 5′ leader or the mutated 5′ leader of *Rh1-RA*, the first 15 bases of the CDS of *Rh1-RA*, the Venus yellow fluorescent protein gene, and the 3′ UTR of *Rh1-RA* were synthesized and cloned into the pBFv-UAS3 plasmid (Addgene #138399). The sequence of the resultant plasmid is provided in the *Supplementary file 3*. The plasmid was then injected into $y^1$ $v^1$ P{nos-phiC31}; P{CaryP}attP40, and their progenies were screened for a *v*+phenotype. A single transformant was crossed to $y^1$ $cho^2$ $v^1$; Sp/CyO balancer to establish a transgenic line.

## Immunohistochemistry and fluorescent in situ hybridization

Immunohistochemistry (*Figure 2A*, *Figure 2—figure supplement 1*) was performed as previously described with minor modifications (*Kanno et al., 2021*). Briefly, dissected male fly brains were fixed in 2% paraformaldehyde in PBS for 1 hr at room temperature, washed three times with PBST (0.1% Triton X-100 in PBS), blocked with 3% goat serum in PBST for 30 min, then incubated with the

primary antibody solution at 4°C overnight (mouse anti-FLAG (1:1000; Sigma-Aldrich; F1804), mouse anti-RpS6 (1:200; Cell Signaling; 54D2), and rat anti-elav (1:20; DSHB; 7E8A10)). Subsequently, the brains were washed three times with PBST, incubated with the secondary antibody solution at 4°C overnight (anti-mouse Alexa Fluor 488 (1:400; Invitrogen; A11001), anti-mouse Cy3 (1:1000; Jackson ImmunoResearch; 115-166-003), and anti-rat Cy3 (1:200; Jackson ImmunoResearch; 112-166-003)), washed three times with PBST, and mounted with 86% glycerol in PBS.

Fluorescent in situ hybridization, combined with immunohistochemistry, was performed in a similar manner to *Yang et al., 2017* with several modifications (*Figure 5E–H*). Dissected male fly brains were fixed in PBS containing 3% formaldehyde, 1% glyoxal, and 0.1% methanol for 30 min at room temperature, followed by three quick washes with PBT (0.5% Triton X-100 in PBS). Consistent with the previous study, addition of glyoxal to the fixative improved the FISH signal (*Yao et al., 2021*). The buffer was then exchanged to the wash solution (10% Hi-Di Formamide [Thermo Fisher Scientific; 4311320] in 2× saline sodium citrate) and was incubated at 37°C for 5 min. Subsequently, the brains were incubated with the custom-made Stellaris *Venus* or *GFP* probes (100 nM; see *Supplementary file 3* for the sequences; LGC BioSearch Technologies) and the primary antibody (mouse anti-Repo [1:100; DSHB; 8D12]) in the hybridization buffer (10% Hi-Di Formamide in hybridization buffer [Stellaris RNA FISH Hybridization Buffer, SMF-HB1-10]) at 37°C for 16 hr. The probes and the antibody were then removed by washing the samples quickly three times with preheated wash solution at 37°C, followed by three washes for 10 min at room temperature. Blocking was performed with 3% normal goat serum in PBT for 30 min at room temperature. The secondary antibody (Cy3 goat anti-mouse [1:2000; Jackson ImmunoResearch; 115-166-003]) was then added and was incubated at 4°C overnight. The samples were washed once quickly, three times for 20 min and once for 60 min with PBT, and then mounted in 86% glycerol in 1× Tris–HCl buffer (pH 7.4).

Tissues to detect native GFP or Venus signals (*Figure 5—figure supplement 1*) were prepared as follows: dissected brains were fixed in PBS containing 3% formaldehyde, 1% glyoxal, and 0.1% methanol for 30 min at room temperature, followed by one quick wash and three washes for 10 min with PBT. The samples were then mounted in 86% glycerol in 1× Tris–HCl buffer (pH7.4).

### Imaging and microscopes

Imaging was done on the Olympus FV1200 confocal microscope with GaAsP sensors. A ×100/1.35 silicone immersion objective (UPLSAPO100XS, Olympus) or ×30/1.05 silicone immersion objective (UPLSAPO30XS) was used. Scan settings were kept constant across specimens to be compared.

### Acknowledgements

We thank Dr. Steven DeLuca (Brandeis University), Dr. Atsushi Sugie (Niigata University), and Dr. Yohei Nitta (Niigata University) for kindly providing the transgenic flies. We also thank Dr. Yusuke Kimura and Dr. Yukihide Tomari (the University of Tokyo) for designing the fly rRNA-depletion probes, Dr. Jasper Janssens (ETH Zurich), Dr. Hongjie Li (Baylor College of Medicine), Dr. Gert Hulselmans (KU Leuven), and Dr. Stein Aerts (KU Leuven) for technical advice regarding analysis of the Fly Cell Atlas data, Ayako Abe (Tohoku University) for technical assistance, Dr. Takashi Makino (Tohoku University) for critical discussion, Madoka Ichinose for critical comments on the graphic design, and the HOKUSAI Sailing-Ship supercomputer facility at RIKEN for computational supports. This study was supported by the Ministry of Education, Culture, Sports, Science and Technology (MEXT): 21K06369 (to TI), 21H05713 (to TI), JP20H05784 (to SI), JP21K15023 (to YS), 22H05481 (to HT), 22KK0106 (to HT), 20H00519 (to HT); Japan Society for the Promotion of Science (JSPS): 21K06369 (to TI), JP21K15023 (to YS); Japan Agency for Medical Research and Development (AMED): JP20gm1410001 (to SI); Takeda Life Science Research Grant (to TI); RIKEN-Tohoku Univ Science & Technology Hub Collaborative Research Program (to TI and YS), 'Biology of Intracellular Environments' (to SI), Special Postdoctoral Researchers (to YS), and Incentive Research Projects (to YS), Tohoku University Research Program 'Frontier Research in Duo' (to HT).

# Additional information

## Competing interests

Hiromu Tanimoto: Reviewing editor, eLife. The other authors declare that no competing interests exist.

## Funding

| Funder | Grant reference number | Author |
|---|---|---|
| Ministry of Education, Culture, Sports, Science and Technology | 21K06369 | Toshiharu Ichinose |
| Ministry of Education, Culture, Sports, Science and Technology | 21H05713 | Toshiharu Ichinose |
| Ministry of Education, Culture, Sports, Science and Technology | JP20H05784 | Shintaro Iwasaki |
| Ministry of Education, Culture, Sports, Science and Technology | JP21K15023 | Yuichi Shichino |
| Ministry of Education, Culture, Sports, Science and Technology | 22H05481 | Hiromu Tanimoto |
| Ministry of Education, Culture, Sports, Science and Technology | 22KK0106 | Hiromu Tanimoto |
| Ministry of Education, Culture, Sports, Science and Technology | 20H00519 | Hiromu Tanimoto |
| Japan Agency for Medical Research and Development | JP20gm1410001 | Shintaro Iwasaki |
| Takeda Science Foundation | | Toshiharu Ichinose |
| RIKEN | Biology of Intracellular Environments | Shintaro Iwasaki |
| RIKEN | Special Postdoctoral Researchers | Yuichi Shichino |
| RIKEN | Incentive Research Projects | Yuichi Shichino |
| Tohoku University Research Program "Frontier Research in Duo" | | Hiromu Tanimoto |
| The Uehara Memorial Foundation | | Toshiharu Ichinose |

The funders had no role in study design, data collection and interpretation, or the decision to submit the work for publication.

## Author contributions

Toshiharu Ichinose, Conceptualization, Data curation, Formal analysis, Supervision, Funding acquisition, Investigation, Writing – original draft, Project administration, Writing – review and editing; Shu Kondo, Conceptualization, Resources, Writing – review and editing; Mai Kanno, Investigation; Yuichi Shichino, Resources, Software, Funding acquisition, Methodology, Writing – review and editing; Mari Mito, Resources, Methodology; Shintaro Iwasaki, Resources, Software, Supervision, Funding acquisition, Methodology, Writing – review and editing; Hiromu Tanimoto, Conceptualization, Supervision, Funding acquisition, Writing – original draft, Project administration, Writing – review and editing

## Author ORCIDs
Toshiharu Ichinose ⓘ https://orcid.org/0000-0002-6845-9403
Shu Kondo ⓘ http://orcid.org/0000-0002-4625-8379
Yuichi Shichino ⓘ https://orcid.org/0000-0002-0093-1185
Shintaro Iwasaki ⓘ https://orcid.org/0000-0001-7724-3754
Hiromu Tanimoto ⓘ http://orcid.org/0000-0001-5880-6064

Reviewer #1 (Public Review): https://doi.org/10.7554/eLife.90713.3.sa1
Reviewer #3 (Public Review): https://doi.org/10.7554/eLife.90713.3.sa2
Author response https://doi.org/10.7554/eLife.90713.3.sa3

---

# Additional files

## Supplementary files
• Supplementary file 1. Reads on all the annotated genes in Ribo-seq and RNA-seq (TPM as a unit) and the calculated translational efficiency (TE).

• Supplementary file 2. Reads on genes included in the Gene Ontology terms shown in *Figure 2*. All the genes showing at least one read in all conditions are included.

• Supplementary file 3. Sequences of the rRNA-depletion oligo, the translation reporters, and the smFISH probes.

• MDAR checklist

## Data availability
The raw sequence data have been deposited in the National Center for Biotechnology Information (NCBI) database with the project code (PRJNA992629). The custom scripts are available in Zenodo (https://doi.org/10.5281/zenodo.10637789).

The following datasets were generated:

| Author(s) | Year | Dataset title | Dataset URL | Database and Identifier |
|---|---|---|---|---|
| Ichinose T, Kondo S, Kanno M, Shichino Y, Mito M, Iwasaki S, Tanimoto H | 2024 | Translatinal regulation enhances distinction of cell types in the nervous system | https://www.ncbi.nlm.nih.gov/bioproject/PRJNA992629 | NCBI BioProject, PRJNA992629 |
| Ichinose T | 2024 | Custom scripts for "Translational regulation enhances distinction of cell types in the nervous system" | https://doi.org/10.5281/zenodo.10637789 | Zenodo, 10.5281/zenodo.10637789 |

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
