## [Editor Report · eLife assessment]

This **valuable** article explores the role of translational regulation in the establishment of differential gene expression between neurons and glia in *Drosophila*. The article uses Ribo-seq to show extensive variation in the translation efficiency of specific transcripts between neurons and glia. The evidence supporting the model is **solid**, although only one example (that exhibits very strong differential transcriptional expression between one class of neurons and glia) is studied in detail for translation efficiency.

---

## [Referee Report · Reviewer #1 (Public Review)]

This study seeks to understand how selective mRNA translation informs cellular identity using the *Drosophila* brain as a model. Using drivers specific for either neurons or glia, the authors express a tagged large ribosomal subunit protein, which they then use as a handle for isolating total mRNA and ribosome footprints. Throughout the study, they compare these data sets to transcriptional and ribosome profiles from the whole fly head, which contains multiple cell types including fat tissue, pigment cells and others, in addition to neurons and glia. Using GO term analyses, they demonstrate the specificity of their cell-type-based ribosome profiling: known glial mRNAs are efficiently translated in glia and likewise in neurons as well. In further examining their RNAseq data set, they find that "neuronal" mRNAs, such as ion channels, are expressed in both neurons and glia, but are translated at higher rates in neurons. Based on this, they hypothesize that neuronal mRNAs are actively suppressed in glia, and next seek to determine the underlying mechanism. By meta-analysis of all mapped ribosome footprints, they find that glia have higher ribosome occupancies in the 5' leader of neuronal mRNAs. This is corroborated by individual ribosome occupancy profiles for several neuronal mRNAs. In 5'leaders containing upstream AUG codons, they find that the glial data sets show an enrichment of ribosomes at these upstream start sites. They thus conclude that that 5' leaders containing upstream AUGs confer translational suppression in glia.

Overall, the sequencing data sets generated in this study and their subsequent bioinformatic analyses seem robust and reliable. Their data echo the trends of cell-type specific translational profiles seen in previous studies (e.g. 27380875, 30650354), and making their data sets and analyses accessible to the broader scientific community would be quite helpful. The findings are presented in a logical and methodical manner, and the data are depicted clearly. The authors' results that 5' leaders facilitate translation suppression is well-supported in literature. However, they overinterpret their data by claiming that such suppression is key for maintaining glial/neuronal identity (it is even featured in their title), but do not present any evidence that loss of such regulation has any impact on cellular identity. In many places, the authors do not acknowledge possible biases in their analytical methods, or consider alternate explanations for their data. These weaken the manuscript in its current form, but many of these issues which I describe below, are rectifiable with modest effort.

(1) The authors' data in Fig. 2-S1A-B shows substantial cell-to-cell variation in RpL3::FLAG expression. The authors do not consider that this variation may cause certain neuronal/glial types to be overrepresented in their datasets. In related, the authors do not discuss whether RpL3::FLAG only present in the cell body or if it is also trafficked to the neuronal/glial processes where localized translation is known to occur (reviewed in 31270476).

(2) The RNA-seq data set that they use to calculate translation efficiency (TE) only represents mRNAs associated with RpL3::FLAG, which is part of the large ribosome subunit. As the authors are likely aware, there are mRNAs on which the full ribosome moiety does not assemble and these are effectively excluded from this data set. Ideally, a more complete picture of the mRNA landscape can be obtained by 40S subunit profiling but I appreciate that this is technically very challenging. At minimum, this caveat needs to be acknowledged.

How does the TPM of differentially regulated transcripts (such as those in Fig. 2H) compare between whole heads, neurons and glia? Since the whole head RNA-seq data was not from an enriched sample, this might serve as a decent proxy for showing that the neuron/glia RNA-seq data sets are representative of RNA abundance.

(3) The analysis in Fig. 2F shows that low abundance mRNAs in glia are further translationally suppressed, which the authors point out in lines 151-152. However, this data also shows that mRNAs with a 1:1 ration in neuron:glia (which fall in the 0.5-1 and 1-2 bin) have a TE-1; this suggests that on average, mRNAs that are equally abundant are translated equally efficiently. This is the opposite of the thesis presented in Fig. 2G-H where many mRNAs of equal abundance in neurons and glia are actually poorly translated in glia. How do the authors reconcile these observations?

It is also unclear from the manuscript whether all mRNAs were considered for the analysis in Fig. 2F or if some cutoff was employed.

(4) Throughout the manuscript the authors favor a "translation suppression" model wherein glia (for example) actively suppress neuronal mRNAs, and this is substantiated in Fig. 3C showing higher ribosome occupancy on 5' leaders than in coding regions. However, they show no evidence that glial mRNAs (such as those indicated in Fig. 2B and 2-S2B) present a different pattern, say that of higher ribosome occupancy in CDS vs. 5' leaders. This type of a positive control is a glaring omission from many of their analyses, including ribosome occupancy at upstream AUG codons (Fig. 4).

In related, to make a broad case (as they do in the title) that differential translation regulation specifies multiple cell types, it is necessary to show the corollary: that glial mRNAs (repo, bnb, pnt, etc) are suppressed in neurons. There is an inkling of this evidence in Fig. 3-S1 where fat body mRNAs in neurons are shown to have low ribosome occupancy in the CDS regions and enhanced occupancy in the 5' leader region. This data is not quantified, nor is a control neuron mRNA shown as a reference for what the ribosome occupancy profile of an actively translated mRNA looks like in a neuron.

(5) The cell-type specific ribosome profiling data sets in the manuscript are from mRNAs associated with 80s subunits that have been treated with cycloheximide during sample preparation. Cycloheximide, and many other translation inhibitors, are known to non-uniformly bias reads towards start codons (PMID: 22056041,22927429). This important caveat and its implications on the start-codon occupancy analysis in Fig. 4 are not acknowledged in the manuscript.

Again, the ideal resolution would be ribosome profiling data set from 40S footprinting or harringtonine-treated samples (PMIDs: 32589966, 27487212, 32589964) to show true accumulation of ribosomes at AUG codons. In the absence of such a data set, a comparative meta-analysis of the ribosome distribution around upstream and initiation AUG codons of differentially translated transcripts from neurons would be a useful control.

(6) The authors chose Rhodopsin 1 (Rh1) as a model mRNA which is translated efficiently in neurons but suppressed in glia. Though the data in Fig. 2-S3B shows higher TE for Rh1 in neurons, the data in 5A show lower ribosome occupancy in the Rh1 CDS in neuron samples (at least in the fragment of the CDS visible). These data are somewhat contradictory.

Further, given that the neuron data are from all nsyb-positive cells but that Rh1 is expressed only in R1-R6 photoreceptors, it is unclear what motivated them to chose Rh1 as opposed to an mRNA that is more broadly expressed in neurons.

(7) Similar to the heterogeneity in nsyb- and repo-GAL4 expression in Fig. 2-S1A-B, Fig. 5C shows substantial variation in the expression of the UAS-GFP reporter driven by tub-GAL4. This variable GAL4 activity makes the mRNA abundance data difficult to interpret. Also, since the authors presume that Rh1 mRNA is expressed in glia (it is not annotated in the RNA-seq analysis in Fig. 2-S2B), would Rh1-GAL4 not be a more apt driver?

These issues are further compounded by the lack of a cellular compartment marker (repo marks glial nuclei) which makes it impossible to determine which cell the mRNA signal is in. There are also no negative controls are presented for the mRNA probes.

Most confoundingly though, the control reporter itself seems to show variable translation efficiencies from one cell to another, with high-GFP protein cells showing lower GFP mRNA and vice versa.

The mRNA:protein ratio may be easier to examine by using repo-GAL4 to specifically drive the Rh1-reporter expression in glia (such as in Fig. 5-S1A) rather than simultaneous expression in both neurons and glia using tub-GAL4.

Comments post revision: The authors have satisfactorily addressed most of my concerns with the study. I appreciate their patient clarification of many of my points, and the revision to text+figures appending more controls. My only minor gripe remains that while their data beautifully show that there is differential regulation of transcripts across neurons and glia, they do not provide evidence that such regulation is required for cell identity. However, I appreciate this is a large experimental ask worthy of another study in and of itself. Overall, I peg this an excellent study that adds substantially to the field of cell-type specific mRNA translation regulation.

---

## [Referee Report · Reviewer #3 (Public Review)]

It is well established that there is extensive post-transcriptional gene regulation in nervous systems, including the fly brain. For example, dynamic regulation of hundreds of genes during photoreceptor development could only be observed at the level of translated mRNAs, but not the entire transcriptomes. The present study instead addresses the role of differential translational regulation between cell types (or rather classes: neurons and glia, as both are still highly heterogenous groups) in the adult fly brain. By performing bulk RNA-seq and Ribo-seq on the same lysates, the authors are able to compare translation efficiency (TE) of all transcripts between neurons and glia. Many genes display differential TE, but interestingly, they tend to be the genes that already show strong differences at their mRNA level. The most striking observation is the finding that neuronal transcripts in glia display increased ribosome stalling at their 5' UTR, and in particular at the start codons of short "upstream ORFs". This could suggest that glia specifically employ a mechanism to upregulate upstream ORF translation, enabling them to better suppress the expression of the genes that have them. And neuronal genes tend to have longer 5' UTRs, perhaps to facilitate this type of regulation.

However, it is difficult to evaluate the functional significance of these differences because the authors provide only one follow-up experiment to their RNA-seq analysis. Venus expressed with the Rh1 UTR sequences may be displaying differential levels between glia and neurons, but I find this image (Fig. 5C) rather unconvincing to support that conclusion. There are no quantifications of colocalization, or even sample size information provided for this experiment. And if there is indeed a difference, it would still be difficult argue this is because of the 5' stalling phenomenon authors observe with Rh1, because they switched both the 5' and 3' UTRs.

I also find it puzzling that the TE differences between the groups are mostly among the transcripts that are already strongly differentially expressed at the transcriptional level. The authors would like to frame this as a mechanism of 'contrast sharpening'; but it is unclear why that would be needed. Rh1, for instance, is not just differentially expressed between neurons and glia, but it is actually only expressed by a very specific neuronal type (photoreceptors). Thus it's not clear to me why the glia would need this 5' stalling mechanism to fully suppress Rh1 expression, while all the other neurons can apparently do so without it.

Response to authors' revisions:

The authors have addressed most of the technical points in their revised manuscript. However, it is still rather unclear whether this mechanism would have any significant impact on differential gene expression between cell types in vivo. Considering that it's mostly occurring on genes that are already strongly differentially transcribed, that doesn't appear very likely.

---

## [Author Response]

The following is the authors’ response to the original reviews.

**Reviewer 3:**
Response to authors' revisions:This reviewer is not convinced that the authors have done enough to satisfactorily address either of the major issues described in the original public review, above.They're still not providing a quantification of Fig. 5D (originally 5C).Their response regarding the expression pattern of Rh1 is particularly concerning, as it represents a misinterpretation of previously published data.The gene encoding Rh1, ninaE, is expressed at such high levels in R1-6 PRs that any RNA-seq data (bulk or single-cell) generated from the optic lobes, no matter what cell-type, will display some ninaE transcripts that are present in the background, as they leak from R1-6 during dissociation steps. This phenomenon has been well described, for instance in Davis et al., 2020, eLife, and in fact led to the development of computational tools to abate such artifacts. In other words: no, *rh1* is not expressed in glia, or any other neuron besides PRs for that matter. Therefore, I remain deeply suspicious about the functional relevance of the regulatory mechanisms described in this paper.

We thank the reviewer for her or his critical comments.

We quantified the cell-type differences in translation of the reporter with *Tub-GAL4* and now show the results in Figure 5F. Consistent with other results, this analysis revealed that the glia-to-neuron ratio of the reporter protein expression is significantly lower when it contains the UTR sequences of *rh1*.

We removed the mRNA counts (former Figure 5A and Figure 5 - figure supplement 1A), as we agree that these may well be contaminated by the very high *rh1* expression in R1-6. We also amended the graph showing the ribosome distribution on the *rh1* mRNA (Figure 5B) to better compare the translational efficiency (footprints normalized with mRNA, in a similar manner to Figure 3C). Now it clearly highlights the cell-type differences of footprint distributions; ribosomes are much more enriched on the CDS (being translated) in neurons, while the fraction of ribosomes on the 5ʹ leader (being stalled) is much higher in glia. We summarized this differential ribosome distribution in a new graph (now Figure 5C).

We apologize for the misleading description of the reporter experiments. Despite the high level of mRNA expression in the R1-6, we chose the 5ʹ leader of *rh1* for the translation reporter, as it contains clear uORFs and differential ribosome accumulation thereon (Figure 5B). This biased ribosome distribution and differential translation are the consistent features for many neuronal genes (Figure 3). We revised the text to clarify this point (Line 195-203).

In summary, we provide more rigorous analysis and extensive revision, which we hope clarified the concern.